# Genetic relatedness of serial rectal isolates of *Acinetobacter baumannii* in an adult intensive care unit of a tertiary hospital in Kuwait

**Ghayda Al-Hashem, Vincent O. Rotimi, M. John Albert**(ID)*

Department of Microbiology, Faculty of Medicine, Kuwait University, Kuwait City, Kuwait

* john@hsc.edu.kw

**Data Availability Statement:** All relevant data are within the paper and its Supporting Information files.

## Abstract

*Acinetobacter baumannii* is an opportunistic pathogen of intensive care unit (ICU) patients. *A. baumannii* colonizes many parts of the body including the gastrointestinal tract. Endemic and epidemic strains are polyclonal. There is no clarity on the origin of polyclonality of *A. baumannii*. The objective of the study was to define the genetic relatedness of serial isolates and the origin of polyclonality. Serial rectal isolates from ICU patients whose rectum was colonized on ≥5 sampling occasions were selected. From a total of 32 eligible colonized patients, isolates from a subgroup of 13 patients (a total of 108 isolates) showing different patterns of colonization as revealed by pulsed-field gel electrophoresis (PFGE) were studied. The isolates were analyzed by PFGE pulsotypes, sequence types (STs) by multi-locus sequence typing (MLST) and clonal complex (CC) by eBURST analysis. Serial isolates constituted a mixture of identical, related and unrelated pulsotypes. Analysis by STs and CCs were less discriminatory. The data suggest a combination of an initial colonizing isolate undergoing mutation as well as colonization by independent isolates. Further clarity on the origin of diversity should be better obtained by whole-genome sequencing.

## Introduction

*Acinetobacter baumannii* causes severe nosocomial infections in critically ill patients and is involved in many hospital outbreaks world-wide. It colonizes skin and mucous membranes including the gastrointestinal tract [1,2]. This organism has the propensity for acquiring multiple resistance genes with phenotypic expression of multidrug-resistant (MDR) characteristics. MDR strains are now endemic in many hospitals around the world, including hospitals in Kuwait [3,4]. Choosing appropriate molecular typing methods is vital for investigating epidemiological lineages of the isolates and for infection control. Numerous molecular typing methods are available including pulsed-field gel electrophoresis (PFGE) [5], random amplified polymorphic DNA (RAPD) analysis [6], ribotyping [7], multilocus PCR and electrospray ionization mass spectrometry (PCR/ESI-MS) [8], amplified fragment length polymorphism (AFLP) analysis [7], repetitive extragenic palindromic sequence-based PCR (rep-PCR) [9], and infrequent-restriction-site analysis [10]. PFGE is used as a common method for typing *A. baumannii* isolates [11]. Even though PFGE has a high discriminatory power, it cannot be

**Funding:** The study was funded by a Kuwait University Research grant (YM03/15) to MJA. The funder had no role in study design, data collection and analysis, decision to publish, or preparation of the manuscript.

**Competing interests:** The authors have declared that no competing interests exist.

used for comparison of data among laboratories because of technical variations [12, 13]. *A. baumannii* has a unique $bla_{OXA-51-like}$ gene that may be used for species identification and PCR-based typing into sequence groups (SGs) [14]. Multilocus sequence typing (MLST), has been used successfully for global comparison of isolates [15,16]. eBURST is used to compare the relatedness of isolates by a single locus difference as PFGE compares the isolates by the size of the restricted segments of DNA. In eBURST analysis, the relationship of isolates is presented graphically [17]. Whole-genome sequencing (WGS) has been used recently for epidemiological investigations [18]. Even though, it is more discriminatory than other methods, the technology is complex and expensive and not amenable to many laboratories [19].

In the adult intensive care unit (ICU) of the Mubarak Al Kabeer Hospital, which is a tertiary teaching hospital in Kuwait, there have been several outbreaks of MDR *A. baumannii* infection [4,20]. The outbreak isolates were found to be multiple clones that were on many occasions not similar or related when typed by PFGE. Also, patient and hospital environmental isolates were not related. Endemic strains from ICUs without outbreaks also exhibited polyclonality [4]. Hence, we hypothesized that the patient gut environment may contribute to the origin of genetic diversity of these isolates, where the isolates may undergo acquisition or loss of specific mobile gene elements or recombination events under the selection pressure of antibiotic exposure during the prolonged hospital stay of patients. *A. baumannii* has a highly plastic genome with the resultant gain or loss of genetic materials [21]. Therefore, we studied the genetic relatedness of serial *A. baumannii* isolates colonizing the rectum of adult ICU patients at Mubarak Al Kabeer Hospital. We typed the isolates by the commonly available PFGE, MLST and eBURST methods to determine whether these methods will give sufficient insights into the evolution of colonizing strains.

## Methodology

### Patients and study design

This study was carried out in the adult ICU of Mubarak Al Kabeer Hospital, Kuwait. The hospital has a total of 850 beds including 30 beds in the adult ICU. The catchment area for this hospital covers a population of approximately 800,000 people. The period of study was from March 2015 to June 2016. Rectal swabs were collected from newly admitted patients on the day of admission, third day after admission and then twice weekly until the patient was either discharged or dead. Patients who had five or more positive cultures on different days were included in the final analysis. Relevant information such as age, gender, nationality, diagnosis and comorbidity, antibiotic therapy, previous hospital admission, and live discharge or death, were carefully recorded.

### Isolation and identification

The rectal swabs were inoculated into an enrichment broth containing acetate and incubated aerobically at 37°C for 48 h [22]. The enriched culture was subcultured onto *Acinetobacter* CHROMagar (CHROMagar, Paris, France) and incubated at 37°C for 48 h. Different morphotypes of typical large red colonies were selected for further identification by API NE20 (bioMérieux, l'Etoile, Marcy, France) and confirmed by a duplex PCR assay for *gyrB* gene according to Higgins et al 2007 [23].

### Antibiotic susceptibility testing

Antibiotic susceptibility testing of the isolates was performed by E-test method (bioMerieux) and interpreted according to Clinical and Laboratory Standards Institute (CLSI) susceptibility

criteria [24]. Susceptibility to tigecycline was determined according to the criteria of Talaga et al [25]. Susceptibility to colistin was performed by agar dilution method and interpreted by the CLSI criteria [24].

## Typing by DiversiLab

To determine how many colonies from a patient culture plate should be analyzed, we hypothesized that there are different colony morphotypes of *A. baumannii* on CHROMagar and each colony morphotype represented a different genetic type. To test this hypothesis, in a preliminary pilot study, we tested colonies from 12 patients. The isolates were typed by repetitive sequence-based PCR (DiversiLab[TM] System; bioMérieux). Clonal relatedness was analyzed with the DiversiLab software using the Pearson correlation statistical method. Relatedness was defined as: ≥98% similarity as identical, ≥ 95% and <98% similarities as related, and <95% similarity as unrelated.

## Typing by pulsed field-gel electrophoresis (PFGE)

PFGE was performed as previously described by Seifert et al 2005 with *Apa*I restriction enzyme. [26]. The apparatus and conditions as [27]. Strain relatedness was analyzed by BioNumerics software (Applied Maths, bioMérieux). The percentage of similarity was calculated by dice coefficient with 1.5% tolerance and 1.5% optimization with a cutoff point of 100% for identical, ≥80% related and <80% unrelated isolates [28,29]. Major pulsotypes were represented by different clades. Isolates within the same clades were denoted as subtypes if they exhibited ≥80% and < 100% relatedness. Strain relatedness as identical, related and unrelated was also determined manually by the criteria of Tenover et al [29].

Patients were given alphabetical identification and the serial isolates from a patient were denoted by the patient alphabet and a number representing the sampling number. For example, serial isolates of patient A were denoted as A1, A2, A3, etc. The relatedness of subsequent isolates to the first isolate was indicated as identical (I), related (R), or unrelated (U). If more than one colony morphotypes were studied, the morphotypes were denoted by lower case alphabets. For example, A5a and A5b, meant that on the 5[th] sampling of patient A, there were two colony morphotypes, a and b.

## Grouping of patients based on PFGE

Based on PFGE typing of serial isolates, patients were grouped based on appearance and disappearance of various PFGE types (Table 1). This analysis segregated patients into 4 groups. Serial isolates from one or more patients representing each group was further studied as outlined below.

**Table 1. Grouping pattern of 270 isolates from 32 patients by PFGE.**

| Group | Relatedness of isolates | Patient (number of isolates) |
|---|---|---|
| 1 | Colonization with identical and related isolates | $E_{(7)}$, $Q_{(5)}$, $Y_{(6)}$ |
| 2 | Colonization with identical, related and unrelated isolates | $A_{(6)}$, $C_{(8)}$, $F_{(5)}$, $K_{(6)}$, $L_{(12)}$, $M_{(10)}$, $N_{(13)}$, $P_{(8)}$, $R_{(8)}$, $S_{(8)}$, $U_{(5)}$, $W_{(9)}$, $Z_{(12)}$, $AA_{(10)}$, $AB_{(12)}$, $AD_{(8)}$, $AF_{(6)}$, |
| 3 | Colonization with related and unrelated isolates | $B_{(7)}$, $D_{(7)}$, $G_{(9)}$, $H_{(6)}$, $O_{(6)}$, $T_{(7)}$, $V_{(5)}$ |
| 4 | Colonization by unrelated isolates | $I_{(12)}$, $J_{(16)}$, $X_{(11)}$, $AC_{(8)}$, $AE_{(12)}$ |

### Multi-locus sequence typing (MLST)

MLST was performed as described previously by Bartual et al, 2005 [30] for the Oxford theme. The final purified product was sequenced in a sequencing machine (3130xl Genetic analyzer, Applied BioSystems, CA, United States). Sequences were trimmed to the required lengths and compared by Clustal X and the sequence type determined on the website https://pubmlst.org/abaumannii/ [31].

### Whole genome sequencing (WGS)

Sequencing libraries were prepared using the Nextera XT DNA sample preparation kit (Illumina, San Diego, CA, USA) and the sequence read data were produced on the Illumina Next-Seq instrument (paired end, 150 base reads). De novo assembly of the read data of the isolate was performed using MegaHit [32]. The resulting draft genome sequence was used to derive MLST (PubMLST: https://pubmlst.org/ for Oxford scheme).

Only the isolate K5 was subjected to WGS because the sequence of *gpi* gene for MLST could not be determined due to lack of priming of the forward primer (See under RESULTS, S3 Fig and S3 Table).

### eBURST analysis

eBURST was used to analyze the MLST data to determine the evolutionary relationships among the isolates. The eBURST diagram was constructed by version 3.0 software (http://eburst.mlst.net/), using all available data from the *A. baumannii* PubMLST database. A complete MLST database was visualized as a single eBURST diagram.

### Ethics statement

The ethical approval for this study was granted by the Ethics Committee, Ministry of Health, State of Kuwait (approval number 112). All patients voluntarily gave written informed consent for rectal swab collection and data collection.

## Results

### Comparison of colony morphotypes with DiversiLab types

The results of the analysis on 12 patients are shown in S1 Table and sample DiversiLab dendrograms in S1 Fig. Studies of three different colonies from five patients (nos. 1, 5, 7, 9, 10) showing a single morphotype revealed that all three colonies were identical by DiversiLab. On the other hand, when colonies of different morphotypes were studied from the remaining seven patients, the colonies were either related or unrelated, but not identical. Based on this observation, single colonies representing each morphotype were studied from the patients from whom serial rectal samples were analyzed.

### Study of patients with serial rectal swab collection

A total of 493 patients were studied over a period of 16 months (from March 2015 to June 2016) from whom 1912 rectal swab specimens were collected. Of these, 117 (23.7%) patients and 475 (24.8%) swabs were positive with red colonies resembling *Acinetobacter* spp. on *Acinetobacter* CHROMAgar. The isolates were then confirmed as *A. baumannii* by $bla_{gyrB}$ PCR assays. Seventy-three (62.4%) patients were colonized after 72 h of admission, and 44 patients (37.6%) were colonized on the day of admission. The latter were regarded as colonization before admission to the ICU and therefore omitted from the analysis because we did not know

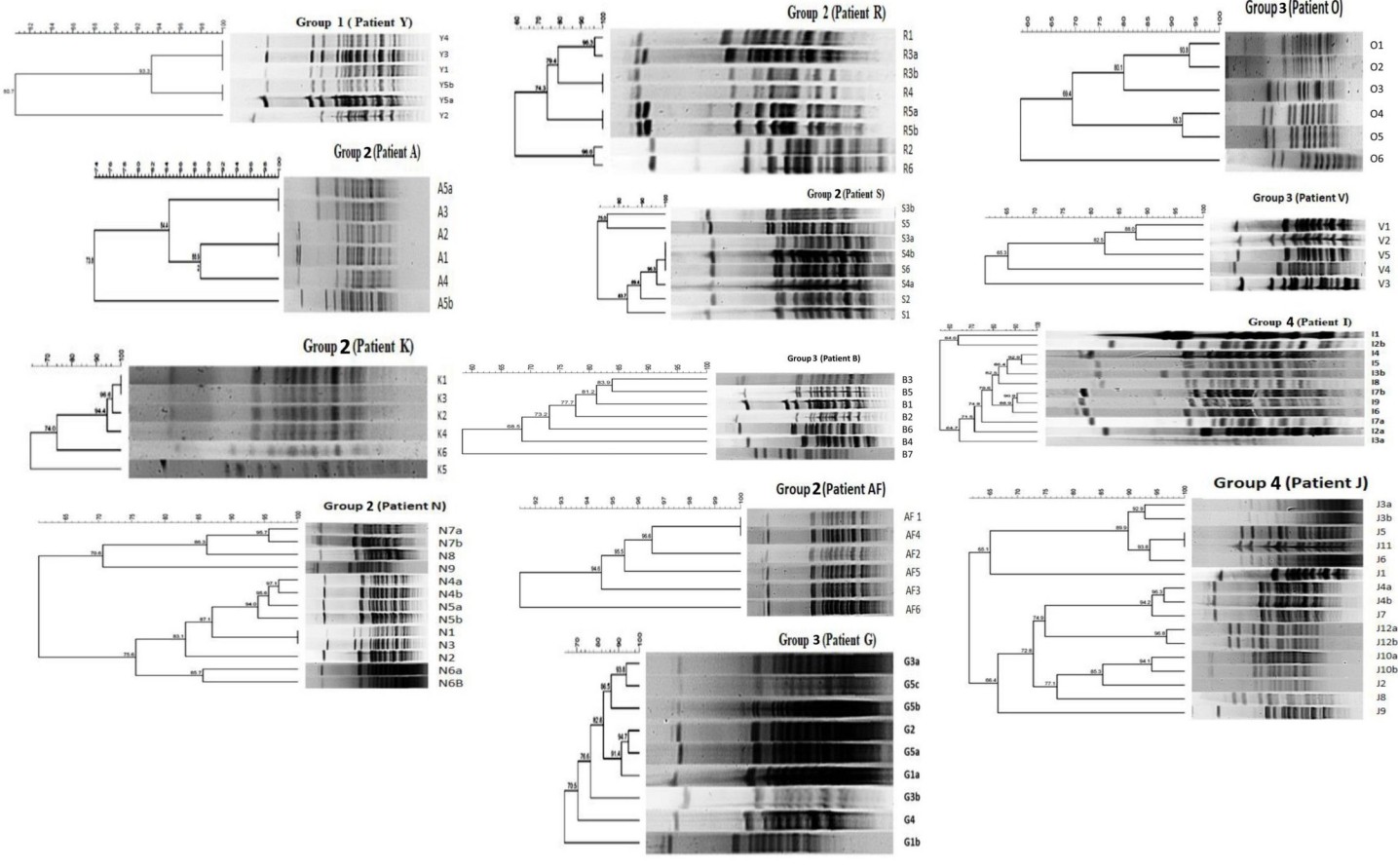

**Fig 1. PFGE dendrograms of thirteen patients (Y, N, R, I, J, S, V, G, O, A, AF, K, B) belonging to six groups determined by PFGE grouping.**

the colonization history of already colonized patients. Of the 73 patients who acquired the isolates in the ICU, 32 (43.8%) were colonized on multiple occasions (≥5 times) yielding a total of 270 isolates.

## Antibiogram of isolates

The antibiotic resistance data are shown in S2 Table. Most of the isolates, 89 (82.4%), were multidrug-resistant (MDR) (resistant to ≥3 antibiotic classes). There was no consistent pattern of resistance in serial isolates from patients.

## Pulse field gel electrophoresis typing

PFGE typing of 270 isolates from 32 patients resulted in the patients being assigned into four groups as shown in Table 1. This grouping is based on the relationship of subsequent isolates to first isolates as I, R or U. Isolates (n = 108) from thirteen patients representing all the four groups (patients Y, N, R, I, J, S, V, G, O, A, AF, K and B) were further studied. The dendrograms of the isolates from these thirteen patients are given in Fig 1 and the relationship of serial isolates are shown in Table 2. There was better differentiation of I and R isolates by BioNumerics method than by Tenover method. Both methods differentiated U isolates similarly.

**Table 2. Typing of serial isolates of *A. baumannii* colonizing the rectum of patients by PFGE, MLST and eBURST.**

| Group | Patient | Isolate no. | Date of isolation | PFGE relatedness by BioNumerics method | PFGE pulsotype by BioNumerics method | PFGE relatedness by Tenover pulsotype | ST no. | eBurst (Clonal complex) |
|---|---|---|---|---|---|---|---|---|
| 1 | Y | Y1 | 19-Feb-2016 | FI | 1a | FI | 195 | CC 208 |
| | | Y2 | 23-Feb-2016 | R | 1b | R | 452 | CC 208 |
| | | Y3 | 2-Mar-2016 | I | 1a | I | 195 | CC 208 |
| | | Y4 | 5-Mar-2016 | I | 1a | I | 195 | CC 208 |
| | | Y5a | 9-Mar-2016 | R | 1c | I | 195 | CC 208 |
| | | Y5b | 9-Mar-2016 | R | 1c | I | 195 | CC 208 |
| 2 | A | A1 | 21-Apr-2016 | FI | 1a | FI | 884 | CC 884 |
| | | A2 | 28-Apr-2016 | I | 1a | I | 884 | CC 884 |
| | | A3 | 19-May-2016 | R | 1b | R | 218 | CC 208 |
| | | A4 | 22-May-2016 | R | 1c | I | 218 | CC 208 |
| | | A5a | 28-May-2016 | R | 1b | R | 884 | CC 884 |
| | | A5b | 28-May-2016 | U | 2 | U | 218 | CC 208 |
| 2 | K | K1 | 19-May-2015 | FI | 1a | FI | 218 | CC 208 |
| | | K2 | 8-Jun-2015 | R | 1b | I | 218 | CC 208 |
| | | K3 | 30-Jun-2015 | I | 1a | I | 218 | CC 208 |
| | | K4 | 24-Jul-2015 | R | 1c | I | 218 | CC 208 |
| | | K5 | 18-Sep-2015 | U | 2 | U | NEW1 | NEW1 |
| | | K6 | 25-Nov-2015 | U | 3 | U | 368 | CC 208 |
| 2 | N | N1 | 3-Nov-2015 | FI | Ia | FI | 195 | CC 208 |
| | | N2 | 6-Nov-2015 | R | 1b | U | 195 | CC 208 |
| | | N3 | 18-Dec-2015 | I | 1a | I | 195 | CC 208 |
| | | N4a | 12-Feb-2016 | R | 1c | R | 195 | CC 208 |
| | | N4b | 12-Feb-2016 | R | 1d | R | 195 | CC 208 |
| | | N5a | 19-Feb-2016 | R | 1e | R | 195 | CC 208 |
| | | N5b | 19-Feb-2016 | R | 1f | R | 195 | CC 208 |
| | | N6a | 23-Feb-2016 | U | 2a | U | 195 | CC 208 |
| | | N6b | 23-Feb-2016 | U | 2b | U | 195 | CC 208 |
| | | N7a | 2-Mar-2016 | U | 3a | U | 195 | CC 208 |
| | | N7b | 2-Mar-2016 | U | 3b | U | 195 | CC 208 |
| | | N8 | 2-Apr-2016 | U | 3c | U | 218 | CC 208 |
| | | N9 | 20-Apr-2016 | U | 4 | U | 218 | CC 208 |
| 2 | R | R1 | 17-Jan-2016 | FI | 1a | FI | 218 | CC 208 |
| | | R2 | 2-Feb-2016 | U | 2a | U | 884 | CC 884 |
| | | R3a | 9-Feb-2016 | R | 1b | R | 884 | CC 884 |
| | | R3b | 9-Feb-2016 | U | 3 | U | NEW2 | NEW2 |
| | | R4 | 19-Feb-2016 | U | 3 | U | NEW2 | NEW2 |
| | | R5a | 9-Mar-2016 | U | 4 | U | 884 | CC 884 |
| | | R5b | 9-Mar-2016 | U | 4 | U | 884 | CC 884 |
| | | R6 | 12-Mar-2016 | U | 2b | U | 884 | CC 884 |
| 2 | S | S1 | 6-Nov-2015 | FI | 1a | FI | 218 | CC 208 |
| | | S2 | 16-Feb-2016 | R | 1b | R | 195 | CC 208 |
| | | S3a | 12-Mar-2016 | R | 1d | R | 195 | CC 208 |
| | | S3b | 12-Mar-2016 | U | 2 | U | 195 | CC 208 |
| | | S4a | 23-Mar-2016 | R | 1c | R | 195 | CC 208 |
| | | S4b | 23-Mar-2016 | R | 1d | R | 195 | CC 208 |
| | | S5 | 13-Apr-2016 | U | 3 | U | 195 | CC 208 |
| | | S6 | 4-May-2016 | U | 1d | R | 195 | CC 208 |

(*Continued*)

**Table 2.** (Continued)

| Group | Patient | Isolate no. | Date of isolation | PFGE relatedness by BioNumerics method | PFGE pulsotypeby BioNumerics method | PFGE relatedness by Tenover pulsotype | ST no. | eBurst (Clonal complex) |
|---|---|---|---|---|---|---|---|---|
| 2 | AF | AF1 | 9-Apr-2016 | FI | 1a | FI | 218 | CC 208 |
| | | AF2 | 16-Apr-2016 | R | 1b | I | 218 | CC 208 |
| | | AF3 | 27-Apr-2016 | R | 1c | I | 218 | CC 208 |
| | | AF4 | 7-May-2016 | I | 1a | I | 218 | CC 208 |
| | | AF5 | 11-May-2016 | R | 1d | I | 218 | CC 208 |
| | | AF6 | 26-May-2016 | U | 2 | I | 218 | CC 208 |
| 3 | B | B1 | 31-Mar-2015 | FI | 1a | FI | 218 | CC 208 |
| | | B2 | 3-Apr-2015 | U | 2 | U | 218 | CC 208 |
| | | B3 | 11-Apr-2015 | R | 1b | U | 218 | CC 208 |
| | | B4 | 21-Apr-2015 | U | 3 | U | 218 | CC 208 |
| | | B5 | 24-Apr-2015 | R | 1c | U | 218 | CC 208 |
| | | B6 | 28-Apr-2015 | U | 4 | U | 1208 | CC 355 |
| | | B7 | 9-May-2015 | U | 5 | U | 218 | CC 208 |
| 3 | G | G1a | 28-Apr-2016 | FI | 1a | FI | 218 | CC 208 |
| | | G1b | 28-Apr-2016 | U | 2 | U | 218 | CC 208 |
| | | G2 | 1-May-2016 | R | 1b | R | 218 | CC 208 |
| | | G3a | 2-Jun-2016 | R | 1c | R | 218 | CC 208 |
| | | G3b | 2-Jun-2016 | U | 3 | U | 218 | CC 208 |
| | | G4 | 8-Jun-2016 | U | 4 | U | 218 | CC 208 |
| | | G5a | 16-Jun-2016 | R | 1d | R | 218 | CC 208 |
| | | G5b | 16-Jun-2016 | R | 1e | R | 218 | CC 208 |
| | | G5c | 16-Jun-2016 | R | 1f | R | 218 | CC 208 |
| 3 | O | O1 | 21-Aug-2015 | FI | 1a | FI | 1980 | CC 1980 |
| | | O2 | 1-Sep-2015 | R | 1b | R | 1980 | CC 1980 |
| | | O3 | 19-Oct-2015 | R | 1c | U | NEW3 | NEW3 |
| | | O4 | 27-Oct-2015 | U | 2a | U | 1980 | CC 1980 |
| | | O5 | 30-Oct-2015 | U | 2b | U | 1980 | CC 1980 |
| | | O6 | 6-Nov-2015 | U | 3 | U | 218 | CC 208 |
| 3 | V | V1 | 8-Jan-2016 | FI | 1a | FI | 218 | CC 208 |
| | | V2 | 12-Jan-2016 | R | 1b | U | 1418 | CC 234 |
| | | V3 | 22-Jan-2016 | U | 2 | U | 1418 | CC 234 |
| | | V4 | 26-Jan-2016 | U | 3 | U | 1418 | CC 234 |
| | | V5 | 2-Feb-2016 | R | 1c | I | 1418 | CC 234 |
| 4 | I | I1 | 12-May-2015 | FI | 1 | FI | NEW4 | NEW4 |
| | | I2a | 22-May-2015 | U | 2 | U | 368 | CC 208 |
| | | I2b | 22-May-2015 | U | 3 | U | 368 | CC 208 |
| | | I3a | 5-Jun-2015 | U | 4 | U | 368 | CC 208 |
| | | I3b | 5-Jun-2015 | U | 5a | U | 368 | CC 208 |
| | | I4 | 4-Sep-2015 | U | 5b | U | 218 | CC 208 |
| | | I5 | 13-Oct-2015 | U | 5c | U | 218 | CC 208 |
| | | I6 | 17-Nov-2015 | U | 6a | U | 884 | CC 884 |
| | | I7a | 19-Feb-2016 | U | 7 | U | 884 | CC 884 |
| | | I7b | 19-Feb-2016 | U | 6b | U | 884 | CC 884 |
| | | I8 | 9-Mar-2016 | U | 5d | U | 195 | CC208 |
| | | I9 | 20-Apr-2016 | U | 6c | U | 884 | CC884 |

*(Continued)*

**Table 2.** (Continued)

| Group | Patient | Isolate no. | Date of isolation | PFGE relatedness by BioNumerics method | PFGE pulsotypeby BioNumerics method | PFGE relatedness by Tenover pulsotype | ST no. | eBurst (Clonal complex) |
|---|---|---|---|---|---|---|---|---|
| 4 | J | J1 | 22-May-2015 | FI | 1 | FI | 218 | CC 208 |
| | | J2 | 2-Jun-2015 | U | 2a | U | 218 | CC 208 |
| | | J3a | 26-Jun-2015 | U | 3a | U | 218 | CC 208 |
| | | J3b | 26-Jun-2015 | U | 3b | U | 218 | CC 208 |
| | | J4a | 3-Jul-2015 | U | 4a | U | 218 | CC 208 |
| | | J4b | 3-Jul-2015 | U | 4b | U | 218 | CC 208 |
| | | J5 | 17-Jul-2015 | U | 3c | U | NEW5 | NEW5 |
| | | J6 | 19-Sep-2015 | U | 3d | U | NEW5 | NEW5 |
| | | J7 | 30-Oct-2015 | U | 4c | U | NEW5 | NEW5 |
| | | J8 | 13-Nov-2015 | U | 5 | U | NEW5 | NEW5 |
| | | J9 | 18-Dec-2015 | U | 6 | U | NEW6 | NEW6 |
| | | J10a | 16-Feb-2016 | U | 2b | U | NEW6 | NEW6 |
| | | J10b | 16-Feb-2016 | U | 2c | U | NEW6 | NEW6 |
| | | J11 | 30-Mar-2016 | U | 3c | U | NEW6 | NEW6 |
| | | J12a | 2-Apr-2016 | U | 7a | U | NEW6 | NEW6 |
| | | J12b | 2-Apr-2016 | U | 7b | U | NEW6 | NEW6 |

FI is first isolate; the relationship of first isolate to subsequent isolates are I identical, R related or U unrelated; ST no: Sequence type number; CC clonal complex., NEW is new sequence type

## Multi-locus sequence typing

The analysis of serial isolates by MLST showed different patterns. There was a single ST in patients G and AF; two STs in patients B, A, V, S, N and Y; a single ST and two novel STs in patient J; two STs and a novel ST in patients O, R and K; and four STs and a novel ST in patient I. The rank order of prevalence of STs were: 42 isolates of ST218, 24 isolates of ST195, 12 isolates of ST884, 6 isolates of novel ST NEW4, 5 isolates of ST368, 4 isolates each of ST1418, ST1980 and novel ST NEW3, 2 isolates of novel ST NEW1, and one isolate each of ST452, ST1208, and novel STs, NEW2, NEW5 and NEW6.

Novel STs, NEW1 to NEW4 had new allele combinations not described in the Oxford scheme. These are shown in S3 Table. Novel ST, NEW3 in patient O had the following alleles: *gltA*(1), *gyrB* (17), *gdhB*(139), *recA*(12), *cpn60* (had a new sequence [S2 Fig]), *gpi*(170), *rpoD* (5). Novel ST, NEW1 in patient K had the following alleles: *gltA*(21), *gyrB*(15), *gdhB*(139), *recA* (12), *cpn60*(23), *gpi* (could not be sequenced by Sanger sequencing due to lack of binding of forward primer, but sequence obtained by Illumina sequencing, see S3 Fig), *rpoD* (4). All these new MLSTs were uploaded onto the Oxford MLST server.

## Clonal complex determination

eBURST analysis of 108 isolates from the thirteen patients is shown Table 2 and in Figs 2 and 3. The clonal complex (CCs) were CC208, CC234, CC355, and CC884. The singleton isolates were CC1980, and NEW1 to NEW6.

## Comparison of isolates by PFGE, ST and CC

Comparison of differentiation of the isolates by the three typing methods is shown in Table 2.

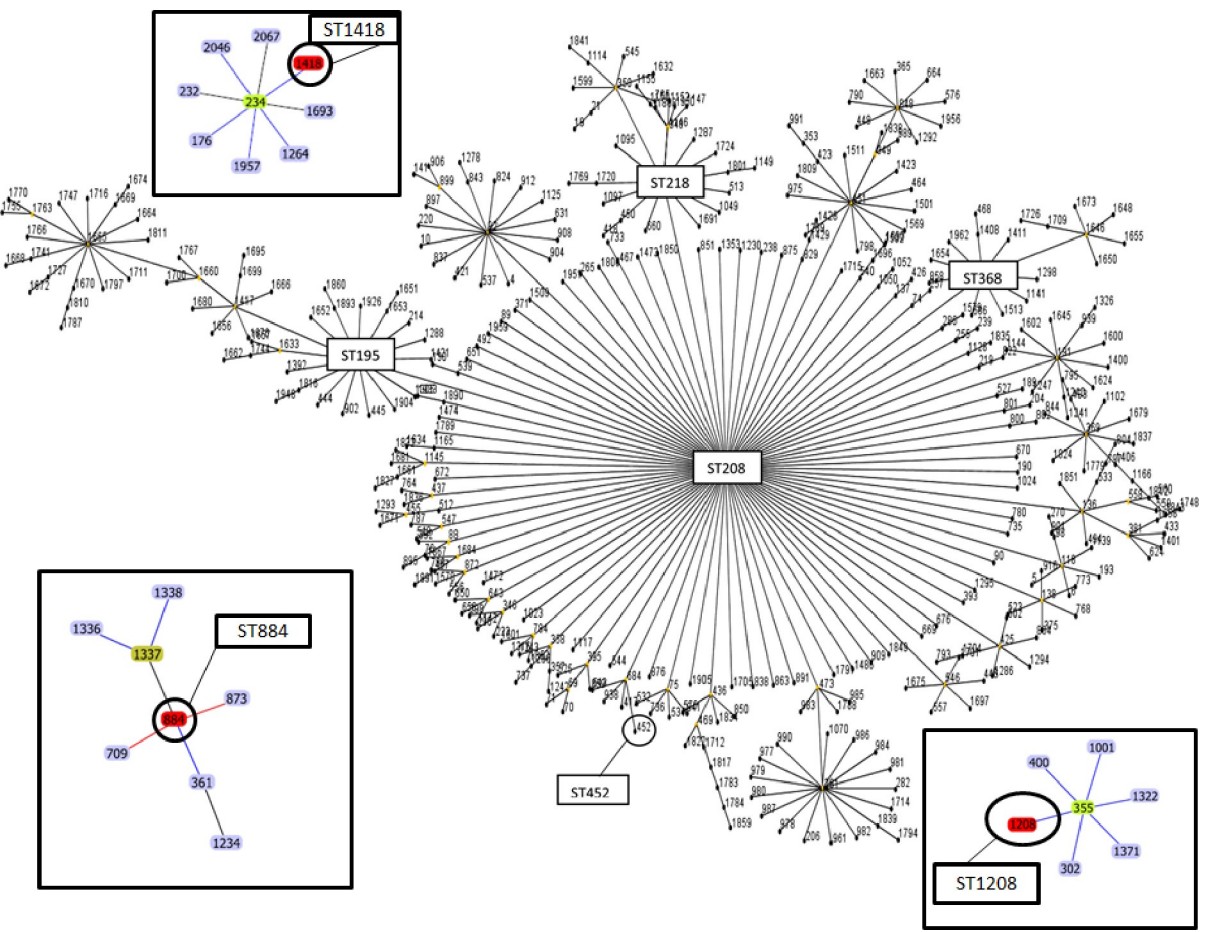

**Fig 2. eBURST diagram generated with MLST data representing the phylogenic relatedness of the seven *A. baumannii* ST types (195, 452, 218,368, 884,1208 and 1418).** ST208, ST884, ST355 and ST234 are the clonal complex origins of CC208, CC234, CC884 and CC355, and the STs close to them differed by a single locus sequence type. Isolates further away have a double or more locus sequence type differences. Seven STs (1980 and 6 novel sequence types) from our study are not shown because they are singletons.

In general, there were more pulsotypes and subtypes by PFGE compared to less number of STs and CCs in all patients. As examples, patient J was colonized by 7 major BioNumerics pulsotypes with 4 pulsotypes showing 3, 4, 3 and 2 subtypes respectively. These isolates belonged to 3 STs and 3 CCs. Patient S was colonized by 3 major BioNumerics pulsotypes with 1 pulsotype showing 4 subtypes. These pulsotypes were represented by 2 STs and 1 CC. These types of better differentiation by pulsotypes can be seen in other patients.

## Discussion

DiversiLab typing was used to ascertain the genetic relationship of colony morphotypes. Our study on 12 patients showed that colonies exhibiting similar morphologies were identical genetically, and colonies of different morphologies differed genetically. Based on these observations, the number of colonies picked for the study of the 32 patients who had serial rectal samples studied, depended on the number of colony morphotypes, that is, one colony representing each morphotype was studied. We chose DiversiLab typing for the study of colony morphotypes because it is an automated method and easier than PFGE. Previous studies have demonstrated that there is a high degree of correlation between DiversiLab typing and PFGE [33].

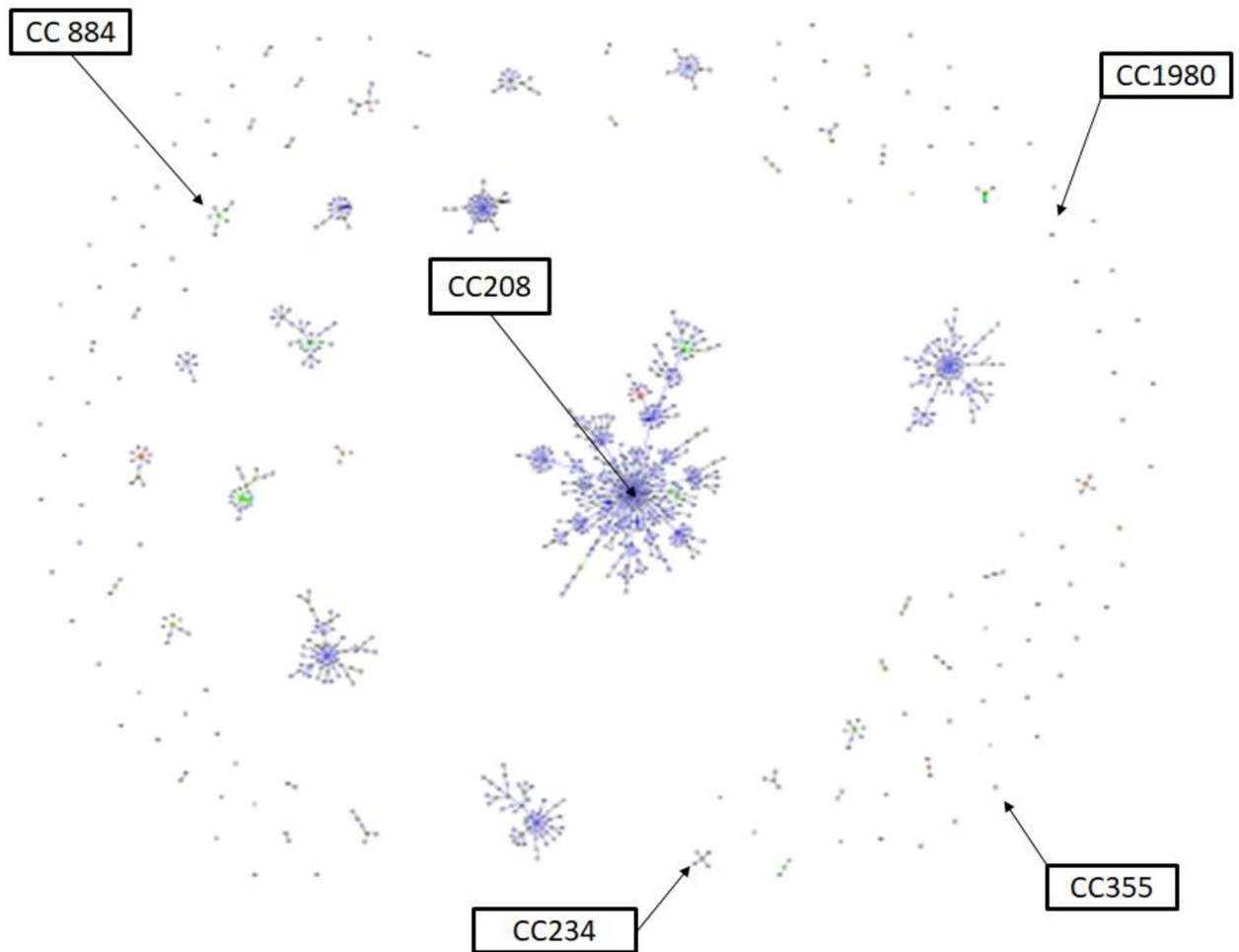

**Fig 3. eBURST diagram generated with MLST data representing phylogenic relatedness of the five major clonal complexes (CC208, CC234, CC884, CC355, CC1980) of *A. baumannii*.**

By simply defining the relationship of the first isolate to the subsequent isolates as I, R or U in PFGE typing, we could assign 32 patients into 4 different groups (Table 1). One or five patients from each group were chosen for the current study. Further analysis of PFGE results into pulsotypes, showed the following grouping of patients: colonization with identical and related isolates (patient Y), colonization with identical, related and unrelated isolates (patients N, R, S, A, AF and K), and colonization with related and unrelated isolates (patients G, J, V, O, B and I). It is tempting to conclude that related isolates may have evolved from an initial isolate that has undergone independent mutation by itself or by genetic exchange with other strains. This is a possibility as *A. baumannii* has a highly plastic genome and is promiscuous in exchange of genetic materials [34]. It is conceivable that unrelated isolates represent independent isolates. Also, the automated BioNumerics pulsotyping has better discrimination than the manual Tenover pulsotyping.

There was less variability among isolates when typed by MLST or eBURST. There were eight STs that were detected in our study, ST218, ST195, ST1208, ST1980, ST452, ST368, ST1418 and ST884. For NEW1 isolate, no amplified product for *gpi* was obtained. This is due to lack of priming of the forward primer (S3 Fig). Others have previously noted a similar

problem with Oxford MLST scheme [35,36]. By CC analysis, most of the isolates belonged to one type, CC208. In addition, there were singletons: ST1980 and those representing the six new STs. Our findings are supported by previous studies that have shown that PFGE typing is more discriminatory than MLST typing or eBURST analysis [37].

It is worth comparing the STs in our study with those from other studies in the region. In an Iranian study [38], the STs were 195, 387, 451, 460 and 848. In a study from Saudi Arabia, eight different STs– 195, 208, 218, 222, 231, 286, 499 and 557- were obtained. In a multicenter study covering the Gulf Cooperation Council (GCC) countries- Saudi Arabia, United Arab Emirates, Sultanate of Oman, Qatar, Bahrain, and Kuwait [39]—seven different STs (195, 208, 229, 436, 450, 452 and 499) and three novel STs were seen. One or two out of eight STs obtained in our study– 195, 218- were present in studies in Iran, Saudi Arabia or GCC countries. Our experience suggests that PFGE typing is a better discriminatory method which is suited for investigation of outbreaks in a hospital, but for inter-country comparison of isolates, STs are suitable even though MLST is less discriminatory.

There are some limitations in our study. First, with regard to isolation of *A. baumannii*, we enriched the rectal swabs in a liquid medium and then subcultured onto a selective agar. It is possible that this procedure might have selected out some strains, but not others. Therefore, the isolation method may not reveal the true picture of colonizing strains. Second, we did not compare the colonizing strains among patients to find out transmission of certain strains between patients. We did not deliberately do this type of comparison as the primary purpose of our study was to characterize the serial isolates colonizing individual patients. Our study was not intended to gauge transmission between patients from the standpoint of infection control.

## Conclusions

Our data suggested that serial colonization of rectum may be due to an initial isolate that has undergone mutation or colonization by independent isolates or a combination of both. Further insight into the origin of isolates colonizing this group of patients in long-stay high dependency units could be obtained by whole-genome sequencing and bioinformatics analysis.

## Supporting information

**S1 Table. The relationship of colonies by DiversiLab dendrograms among similar and different morphotypes.**
(DOCX)

**S2 Table. Antimicrobial susceptibilities of 108 serial rectal *A. baumannii* isolates from 13 patients.**
(DOCX)

**S3 Table. Combination of gene alleles for novel MLSTs.**
(DOCX)

**S1 Fig. The relationship of colonies of similar and different colony morphotypes by DiversiLab dendrogram.** Patients 1, 5 and 7 each had similar colony morphotypes. Three colonies each from these patients were genetically identical by DiversiLab. Patients 3, 6 and 8 had 3, 4 and 6 colony morphotypes, respectively. DiversiLab analysis of these colonies showed different genetic types.
(TIF)

**S2 Fig. Novel *cpn60* allele and genome sequence based MLST type for the Oxford scheme from isolate O3 (designated as NEW3) (fasta file).**
(TIF)

**S3 Fig. *gpi* sequence of *A. baumannii* isolate K5 (designated as NEW4) showing lack of binding of forward primer.**
(TIF)

## Acknowledgments

The authors are grateful to Dieter Bulach, Peter Doherty Institute for Infection and Immunity, the University of Melbourne, Australia, for help with bioinformatics of whole genome sequence.

## Author Contributions

**Conceptualization:** Vincent O. Rotimi, M. John Albert.

**Funding acquisition:** M. John Albert.

**Investigation:** Ghayda Al-Hashem.

**Methodology:** Ghayda Al-Hashem, M. John Albert.

**Project administration:** M. John Albert.

**Resources:** M. John Albert.

**Supervision:** Vincent O. Rotimi, M. John Albert.

**Validation:** M. John Albert.

**Writing – original draft:** Ghayda Al-Hashem.

**Writing – review & editing:** Vincent O. Rotimi, M. John Albert.

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
