## [Decision Letter · Decision Letter 0]

21 Jan 2020

PONE-D-19-35790

Genetic relatedness of serial rectal isolates of Acinetobacter baumannii in an adult
intensive care unit of a tertiary hospital in Kuwait

PLOS ONE

Dear Dr Albert,

Thank you for submitting your manuscript to PLOS ONE. After careful consideration, we
feel that it has merit but does not fully meet PLOS ONE’s publication criteria as it
currently stands. Therefore, we invite you to submit a revised version of the
manuscript that addresses the points raised during the review process.

ACADEMIC EDITOR:   

Dear
Authors, I was in doubt for my decision, cause there are many criticism
to correct. There
are conflicts between the reviews, actually I think the reviewers are
expressing the same concepts; the weak statistical method doesn't permit
this manuscript to be accepted as it is. It needs a very accurate
revision to be published.  Please shorten introduction, materials and methods. Answer to all the
criticism moved by the reviewers in order to make the manuscript ready
for publication. 

We would appreciate receiving your revised manuscript by feb 11th. When you are ready
to submit your revision, log on to https://www.editorialmanager.com/pone/ and select the 'Submissions
Needing Revision' folder to locate your manuscript file.

If you would like to make changes to your financial disclosure, please include your
updated statement in your cover letter.

To enhance the reproducibility of your results, we recommend that if applicable you
deposit your laboratory protocols in protocols.io, where a protocol can be assigned
its own identifier (DOI) such that it can be cited independently in the future. For
instructions see: http://journals.plos.org/plosone/s/submission-guidelines#loc-laboratory-protocols

We look forward to receiving your revised manuscript.

Kind regards,

Martina Crivellari

Academic Editor

PLOS ONE

Journal Requirements:

a) Please provide an amended Funding Statement that declares *all* the funding or
sources of support received during this specific study (whether external or internal
to your organization) as detailed online in our guide for authors at http://journals.plos.org/plosone/s/submit-now.

b) Please state what role the funders took in the study.  If any authors received a
salary from any of your funders, please state which authors and which funder. If the
funders had no role, please state: "The funders had no role in study design, data
collection and analysis, decision to publish, or preparation of the manuscript."

Reviewers' comments:

Reviewer's Responses to Questions

**Comments to the Author**

1. Is the manuscript technically sound, and do the data support the conclusions?

Reviewer #1: Yes

Reviewer #2: Yes

2. Has the statistical analysis been performed
appropriately and rigorously? 

Reviewer #1: Yes

Reviewer #2: Yes

3. Have the authors made all data underlying the
findings in their manuscript fully available?

Reviewer #1: Yes

Reviewer #2: Yes

4. Is the manuscript presented in an intelligible
fashion and written in standard English?

Reviewer #1: Yes

Reviewer #2: Yes

5. Review Comments to the Author

Reviewer #1: The manuscript describes the molecular typing analysis of Acinetobacter
baumannii isolates from a Kuwaitian hospital. It is well written in most parts and
could be interesting for the readers of PLOS One. However, due to undeniable
weaknesses in methodological procedures and data interpretation, it can not be
accepted for publication in its current form.

Major comments:

- the introduction is overlong and should be shortened. This can easily done by a
less detailed explanation of the different typing methods.

- the material and methods part is overlong and could easily be shortened by using
citations (e.g. description of MLST).

- line 50: nowadays, I would not agree that PFGE is still the gold standard. Since
NGS-based typing has become widespread in many countries and as this technique has a
at least equivalent or in most cases higher discriminatory power, it should be named
as the gold standard. However, no consensus criteria for e.g. cgMLST exist, what is
the advantage of PFGE (Tenover criteria).

- line 52: This is not correct. PFGE has a high discriminatory power, even for
isolates from different geographic regions. The real problem is, that PFGE results
are not comparable from one laboratory to another due to technical variations. But
one single laboratory could easily compare isolates from different regions. This has
to be changed.

- the authors did not use the Tenover criteria for outbreak investigation by PFGE. I
of course know that the Tenover criteria should be used only for isolates within
defined time periods, but it would be interesting to know if and how the results
change when interpreted with these consensus criteria. This must be done.

- the band patterns shown in figure 1 are of extremely low quality for some isolates
that in my oppinion do not allow a reliable data interpretation. e.g. the band
pattern isolate Y5a is completely unuseable due to crooked bands. When looking at
the bands in detail, for me it does not look like this is another pulsotype than
Y5b. Additionally, the DNA amounts seem to be very different for different isolates,
which complicates the analysis even more.

The same is the case for isolates R1, R3a, J11, V2 and others... The authors should
reperform the PFGE for isolates with poor band patterns.

- table 2: prior to publication, the new MLSTs and CCs must be numbered in
coordination with the A. baumannii MLST website.

Minor comments:

- line 44: delete "typing of the isolates to determine their relatedness" as this
makes no sense here and seems to be a copy-paste issue

- lines 117/118: "%" is missing for some numbers

- line 121: change it to "cultures...were typed by PFGE."

- line 188: "eBU_R_ST"

Reviewer #2: Al-Hashem et al. present results of a molecular surveillance study on
Acinetobacter baumannii on an adult intensive care unit in a tertiary care centre in
Kuwait. The study was conducted in a setting where A. baumannii is endemic from
March 2016 to June 2017.

The surveillance was based on rectal colonization. From each specimen isolates with
different morphotypes were picked and subsequently genotyped. In a pilot study the
authors analyzed the association of morphotype and genotype in 12 patients. Analyses
were performed with DiversiLab. Based on the pilot study data, the authors concluded
that each morphotype represents one genotype.

Rectal colonization with A. baumannii was studied in 493 patients. In 73 out of 493
patients A. baumannii was detected after 72 h of admission. 32 out of theses 73
patients were positive on more than five occasions (serial isolates). Only patients
with hospital-acquired (> 72 h after admission) and with serial isolates were
included. The authors grouped these 32 patients in six groups based on the
“colonization pattern” and picked 13 patients (2-3 from each group, 108 isolates)
for further genotypic analysis (PFGE, MLST, eBURST).

The key message is the high diversity of hospital-acquired A. baumannii strains
within one patient.

Although the authors did not use whole genome sequencing for genotyping they show
with several other methods (PFGE, MLST) that patients are colonized with several
different strains at single time points and during hospitalization (overtime).

The article is well-written and easy to understand. Nevertheless, there are several
aspects that reduce the scientific impact of this manuscript:

Major revisions

1. The authors do not mention any antibiotic susceptibility data. I recommend adding
this data as it is of interest to the reader if patients are colonized with MDR or
susceptible strains.

2. The authors compared isolates from one patient and not between the patients in the
genotyping analysis. From an infections control perspective it is important to know
if there were any transmissions between the patients. Transmissions can be suspected
if two patients hospitalized at the same time on the same ward acquired isolates
with the same pulsotype/MLST-type (person-to-person- transmission).
Environment-to-patient transmissions are more difficult to prove, especially
retrospectively, however A. baumannii is known to colonize the environment. For
example, PFGE patterns of isolates N3 and Y4 look similar and both patients were
hospitalized during the same time period (end of 2016 and beginning of 2017). Please
explain, why you decided not to compare isolates of different patients and mention
it in the text (limitations?).

3. The discussion is short. Please mention more aspects (epidemiology, are there
similar MLST types in the region, consequences for IPC outbreak control and typing
etc.).

No limitations are mentioned. One limitation is that the sensitivity and specificity
of the microbiological sampling method to detect A. baumannii is not known. Patients
may still be colonized by the first strain overtime even if it is not detected.

4. Line 90 “There was no outbreak during the study period.” Please explain. What kind
of outbreak do you mean? Outbreaks with any kind of bacteria or A. baumannii? Is
there any active surveillance system in place to come to this conclusion? Please,
mention in the text.

5. Isolates were considered as identical (100%), related (99-80%) or unrelated
(<80%) in the PFGE analysis (Line 137-140). In my opinion, this is a very
conservative approach. Even if you run the same isolate on one gel in several lanes
you do not necessarily get 100% similarity. I would suggest: 100-97.5% (highly
related), 97.5-80% related, <80% unrelated. You also chose a less conservative
approach in the DiversLab analysis.

6. Table 1 is part of the results and not part of the methods. Table 1 shows the
different groups based on colonization patterns of the patients overtime (identical,
related, unrelated isolates). The six groups are complex and difficult to
understand.

For example, Patient N is in group 1 “The first isolate disappears and is replaced by
an identical or related isolate over time”. However, in my opinion, patient N
belongs to group 3 “the first isolate disappears and is replaced by related and
unrelated isolates” as there is a relatedness of “FI, -> R-> I -> R->
U”.

A better and more practical subgrouping is proposed in the discussion (line 278-280):
“colonization with identical and related isolates (patient Y), colonization with
identical, related and unrelated isolates (patients N, R, S, A, AF and K), and
colonization with related and unrelated isolates (patients G, J, V, O, B and
I).”.

This is the key message of the manuscript. One must consider that patients are
colonized with several geno- and phenotypes over time, which is important to know in
outbreak situations and to important to trace transmissions. In an outbreak you have
to include several isolates from each patient in the genotyping analysis.

7. Whole genome sequencing (line 180-185): Please mention how many isolates and which
isolates were analyzed by WGS and why.

8. 493 patients were screened over a period of 16 months. Please mention how many
patients were excluded from the study and why (no consent?). I imagine that during
the period more patients than 493 were admitted on the ICU.

9. Why did you exclude patients with A. baumannii present on admission (line 218)?
Please explain.

Minor revisions

Line 42 – 44: Please rephrase. “This organism has the propensity for acquiring
multiple resistance genes with phenotypic expression of multidrug-resistant (MDR)
characteristics. MDR strains are now endemic in many hospitals around the world,
including hospitals in Kuwait [3,4] typing of the isolates to determine their
relatedness.”

Line 47 [6] Consider to mention a study where RAPD was used for A. baumannii.

Line 68. Please mention recent publications where WGS was used for A. baumannii.
There are also several recent publications where a cgMLST scheme was established and
used for A. baumannii.

Lines 117 – 118: Add %.

Line 125 “v/v sarkosyl, pH 7.5),” Remove the bracket.

Line 178: Please mention the website.

Line 200 – 203: Please shift to methods.

Line 245 “These are shown Table S2”. Please rephrase

Line 244 Please mention in the results section that the new MLST types were uploaded
to the MLST server and not in the discussion section (lines 288-289).

Lines 466-474: How many isolates were included in Figure 2 and Figure 3. Please
mention in the text.

6. PLOS authors have the option to publish the peer
review history of their article (what does this mean?). If published, this will
include your full peer review and any attached files.

If you choose “no”, your identity will remain anonymous but your review may still be
made public.

**Do you want your identity to be public for this peer review?** For
information about this choice, including consent withdrawal, please see our
Privacy Policy.

Reviewer #1: No

Reviewer #2: No

---

## [Author Response · Author response to Decision Letter 0]

10 Mar 2020

Response to Reviewers

11/2/2020

Dear Dr. Crivelleri:

Re: PONE-D-19-35790 entitled “Genetic relatedness of serial rectal isolates of
Acinetobacter baumannii in an adult intensive care unit of a tertiary hospital in
Kuwait”

Thank you for forwarding the reviewers’ comments and your comments. My responses are
as below. 

ACADEMIC EDITOR: 

• Dear Authors, I was in doubt for my decision, because there are many criticisms to
correct. 

• There are conflicts between the reviews, actually I think the reviewers are
expressing the same concepts; the weak statistical method doesn't permit this
manuscript to be accepted as it is. It needs a very accurate revision to be
published. 

• Please shorten introduction, materials and methods. Answer to all the criticism
moved by the reviewers in order to make the manuscript ready for publication. 

Thank you for your decision to invite a revised manuscript in spite of many comments
from the reviewers. We have made extraordinary efforts to answer the comments
including shortening the Introduction and Materials & Methods.

Reviewer #1: 

The manuscript describes the molecular typing analysis of Acinetobacter baumannii
isolates from a Kuwaitian hospital. It is well written in most parts and could be
interesting for the readers of PLOS One. However, due to undeniable weaknesses in
methodological procedures and data interpretation, it cannot be accepted for
publication in its current form.

Major comments:

- the introduction is overlong and should be shortened. This can be easily done by a
less detailed explanation of the different typing methods.

Introduction is much shortened now (L39-L74).

- the material and methods part is overlong and could easily be shortened by using
citations (e.g. description of MLST).

Materials and Methods section is much shortened now (L76-L157).

- line 50: nowadays, I would not agree that PFGE is still the gold standard. Since
NGS-based typing has become widespread in many countries and as this technique has a
at least equivalent or in most cases higher discriminatory power, it should be named
as the gold standard. However, no consensus criteria for e.g. cgMLST exist, what is
the advantage of PFGE (Tenover criteria).

This statement is modified now (L50). 

- line 52: This is not correct. PFGE has a high discriminatory power, even for
isolates from different geographic regions. The real problem is, that PFGE results
are not comparable from one laboratory to another due to technical variations. But
one single laboratory could easily compare isolates from different regions. This has
to be changed.

This statement is changed now (L50-L52).

- the authors did not use the Tenover criteria for outbreak investigation by PFGE. I
of course know that the Tenover criteria should be used only for isolates within
defined time periods, but it would be interesting to know if and how the results
change when interpreted with these consensus criteria. This must be done.

We did not investigate outbreak in our study. However, we used Tenover criteria, in
addition, to interpret the relatedness of isolates (L119-L120; Table 2; L196-L197;
L258-L260).

- the band patterns shown in figure 1 are of extremely low quality for some isolates
that in my opinion do not allow a reliable data interpretation. e.g. the band
pattern isolate Y5a is completely unusable due to crooked bands. When looking at the
bands in detail, for me it does not look like this is another pulsotype than Y5b.
Additionally, the DNA amounts seem to be very different for different isolates,
which complicates the analysis even more.

The same is the case for isolates R1, R3a, J11, V2 and others... The authors should
reperform the PFGE for isolates with poor band patterns.

We have repeated PFGE of these isolates several times to improve the quality of gels.
What we have presented are the best patterns we have. I am afraid redoing the gels
is not going to change the quality of gels or conclusions. For Y5a, even though the
middle of the band is a bit drawn up, the pattern is readable. There is a
misunderstanding here. We agree that Y5a and Y5b are of the same pulsotype (1c in
Table 2). With regard to R1 and R3a, they are clearly different with R3a having an
additional band at the top. In Table 2, these isolates are correctly labelled as 1a
and 1b. 

- table 2: prior to publication, the new MLSTs and CCs must be numbered in
coordination with the A. baumannii MLST website.

Unfortunately, in spite of repeated reminders, we have not heard from the curator of
the Oxford MLST scheme. Colleagues in Australia and the UK too have complained of a
lack of response. Important thing is that we have presented detailed information on
the novel STs. If the manuscript is accepted for publication in PLOS ONE, we will
communicate the new information to the journal as and when it becomes available. 

Minor comments:

- line 44: delete "typing of the isolates to determine their relatedness" as this
makes no sense here and seems to be a copy-paste issue

Sorry for the mistake. True, it was a copy-paste issue. The sentence is deleted now
(L44).

- lines 117/118: "%" is missing for some numbers

- line 121: change it to "cultures...were typed by PFGE."

Texts containing these lines are deleted while shortening the Introduction, and
Materials & Methods.

- line 188: "eBU_R_ST"

The spelling mistake is corrected now (L155).

Reviewer #2: Al-Hashem et al. present results of a molecular surveillance study on
Acinetobacter baumannii on an adult intensive care unit in a tertiary care centre in
Kuwait. The study was conducted in a setting where A. baumannii is endemic from
March 2016 to June 2017.

The surveillance was based on rectal colonization. From each specimen isolates with
different morphotypes were picked and subsequently genotyped. In a pilot study the
authors analyzed the association of morphotype and genotype in 12 patients. Analyses
were performed with DiversiLab. Based on the pilot study data, the authors concluded
that each morphotype represents one genotype.

Rectal colonization with A. baumannii was studied in 493 patients. In 73 out of 493
patients A. baumannii was detected after 72 h of admission. 32 out of theses 73
patients were positive on more than five occasions (serial isolates). Only patients
with hospital-acquired (> 72 h after admission) and with serial isolates were
included. The authors grouped these 32 patients in six groups based on the
“colonization pattern” and picked 13 patients (2-3 from each group, 108 isolates)
for further genotypic analysis (PFGE, MLST, eBURST).

The key message is the high diversity of hospital-acquired A. baumannii strains
within one patient.

Although the authors did not use whole genome sequencing for genotyping they show
with several other methods (PFGE, MLST) that patients are colonized with several
different strains at single time points and during hospitalization (overtime).

The article is well-written and easy to understand. Nevertheless, there are several
aspects that reduce the scientific impact of this manuscript:

Major revisions

1. The authors do not mention any antibiotic susceptibility data. I recommend adding
this data as it is of interest to the reader if patients are colonized with MDR or
susceptible strains.

Susceptibility data are now added (L96-L101; L185-L188).

2. The authors compared isolates from one patient and not between the patients in the
genotyping analysis. From an infections control perspective, it is important to know
if there were any transmissions between the patients. Transmissions can be suspected
if two patients hospitalized at the same time on the same ward acquired isolates
with the same pulsotype/MLST-type (person-to-person- transmission).
Environment-to-patient transmissions are more difficult to prove, especially
retrospectively, however A. baumannii is known to colonize the environment. For
example, PFGE patterns of isolates N3 and Y4 look similar and both patients were
hospitalized during the same time period (end of 2016 and beginning of 2017). Please
explain, why you decided not to compare isolates of different patients and mention
it in the text (limitations?).

This is addressed (L284-L288).

3. The discussion is short. Please mention more aspects (epidemiology, are there
similar MLST types in the region, consequences for IPC outbreak control and typing
etc.).

This is now done (L271-L280).

No limitations are mentioned. One limitation is that the sensitivity and specificity
of the microbiological sampling method to detect A. baumannii is not known. Patients
may still be colonized by the first strain overtime even if it is not detected.

Limitations are mentioned now (L281-L288).

4. Line 90 “There was no outbreak during the study period.” Please explain. What kind
of outbreak do you mean? Outbreaks with any kind of bacteria or A. baumannii? Is
there any active surveillance system in place to come to this conclusion? Please,
mention in the text.

Sorry, this statement was included inadvertently. There is no active surveillance,
but if there is a suspicion of outbreak due to any bacteria, then, investigation is
carried out.That way, there was no separate investigation during our study period. 

5. Isolates were considered as identical (100%), related (99-80%) or unrelated
(<80%) in the PFGE analysis (Line 137-140). In my opinion, this is a very
conservative approach. Even if you run the same isolate on one gel in several lanes
you do not necessarily get 100% similarity. I would suggest: 100-97.5% (highly
related), 97.5-80% related, <80% unrelated. You also chose a less conservative
approach in the DiversLab analysis.

Because of the nature of the question asked in the study, it was necessary to adopt
conservative criteria for PFGE analysis. It was a different question with regard to
DiversiLab analysis. Therefore, I am afraid, the adopted approaches are
justified.

6. Table 1 is part of the results and not part of the methods. Table 1 shows the
different groups based on colonization patterns of the patients overtime (identical,
related, unrelated isolates). The six groups are complex and difficult to
understand.

For example, Patient N is in group 1 “The first isolate disappears and is replaced by
an identical or related isolate over time”. However, in my opinion, patient N
belongs to group 3 “the first isolate disappears and is replaced by related and
unrelated isolates” as there is a relatedness of “FI, -> R-> I -> R->
U”.

A better and more practical subgrouping is proposed in the discussion (line 278-280):
“colonization with identical and related isolates (patient Y), colonization with
identical, related and unrelated isolates (patients N, R, S, A, AF and K), and
colonization with related and unrelated isolates (patients G, J, V, O, B and
I).”.

This is the key message of the manuscript. One must consider that patients are
colonized with several geno- and phenotypes over time, which is important to know in
outbreak situations and to important to trace transmissions. In an outbreak you have
to include several isolates from each patient in the genotyping analysis.

We agree with this excellent suggestion and changed the grouping of patients (Table
1in Results; L199-L201).

7. Whole genome sequencing (line 180-185): Please mention how many isolates and which
isolates were analyzed by WGS and why.

This is now mentioned (L149-L151).

8. 493 patients were screened over a period of 16 months. Please mention how many
patients were excluded from the study and why (no consent?). I imagine that during
the period more patients than 493 were admitted on the ICU.

We included all patients.

9. Why did you exclude patients with A. baumannii present on admission (line 218)?
Please explain.

This is now explained (L180-L182).

Minor revisions

Line 42 – 44: Please rephrase. “This organism has the propensity for acquiring
multiple resistance genes with phenotypic expression of multidrug-resistant (MDR)
characteristics. MDR strains are now endemic in many hospitals around the world,
including hospitals in Kuwait [3,4] typing of the isolates to determine their
relatedness.”

Sorry for the mistake. The last line was inadvertently introduced by mistake while
cutting and pasting. It is now deleted (L42).

Line 47 [6] Consider to mention a study where RAPD was used for A. baumannii.

Reference 6 is now for A. baumannii.

Line 68. Please mention recent publications where WGS was used for A. baumannii.
There are also several recent publications where a cgMLST scheme was established and
used for A. baumannii.

Now reference 18 is appropriate (L58).

Lines 117 – 118: Add %.

Line 125 “v/v sarkosyl, pH 7.5),” Remove the bracket.

These are deleted while shortening the Methods section.

Line 178: Please mention the website.

Now mentioned (L141).

Line 200 – 203: Please shift to methods.

Done (L104-L106)).

Line 245 “These are shown Table S2”. Please rephrase

Done (L166, now Table S1).

Line 244 Please mention in the results section that the new MLST types were uploaded
to the MLST server and not in the discussion section (lines 288-289).

Done (L222-L223).

Lines 466-474: How many isolates were included in Figure 2 and Figure 3. Please
mention in the text.

The numbers are not relevant. The figures are constructed based on sequence
types.

I hope that I have answered all comments satisfactorily.

With kind regards

Yours sincerely

Professor M. John Albert

Department of Microbiology

Faculty of Medicine

Kuwait University

Kuwait

---

## [Editor Report · Decision Letter 1]

13 Mar 2020

Genetic relatedness of serial rectal isolates of Acinetobacter baumannii in an adult
intensive care unit of a tertiary hospital in Kuwait

PONE-D-19-35790R1

Dear Dr. Albert,

We are pleased to inform you that your manuscript has been judged scientifically
suitable for publication and will be formally accepted for publication once it
complies with all outstanding technical requirements.

With kind regards,

Martina Crivellari

Academic Editor

PLOS ONE
---

## [Editor Report · Acceptance letter]

20 Mar 2020

PONE-D-19-35790R1 

Genetic relatedness of serial rectal isolates of *Acinetobacter
baumannii* in an adult intensive care unit of a tertiary hospital in
Kuwait 

Dear Dr. Albert:

I am pleased to inform you that your manuscript has been deemed suitable for
publication in PLOS ONE. Congratulations! Your manuscript is now with our production
department. 

With kind regards,

on behalf of

Dr. Martina Crivellari 

Academic Editor

PLOS ONE